# Effect of Myricetin on Lipid Metabolism in Primary Calf Hepatocytes Challenged with Long-Chain Fatty Acids

**DOI:** 10.3390/metabo12111071

**Published:** 2022-11-05

**Authors:** Wei Yang, Mingmao Yang, Yan Tian, Qianming Jiang, Juan J. Loor, Jie Cao, Shuang Wang, Changhong Gao, Wenwen Fan, Bingbing Zhang, Chuang Xu

**Affiliations:** 1College of Animal Science and Veterinary Medicine, Heilongjiang Bayi Agricultural University, Daqing 163319, China; 2Key Laboratory of Animal Biotechnology of the Ministry of Agriculture, College of Veterinary Medicine, Northwest A & F University, Xianyang 712100, China; 3Department of Animal Sciences, Division of Nutritional Sciences, University of Illinois, Urbana, IL 61801, USA; 4College of Veterinary Medicine, China Agricultural University, Beijing 100193, China; 5College of Life Science and Technology, Heilongjiang Bayi Agricultural University, Daqing 163319, China

**Keywords:** myricetin, primary calf hepatocytes, oxidative stress, fatty acid metabolism

## Abstract

Triacylglycerol (TAG) accumulation and oxidative damage in hepatocytes induced by high circulating concentrations of fatty acids (FA) are common after calving. In order to clarify the role of myricetin on lipid metabolism in hepatocytes when FA metabolism increases markedly, we performed in vitro analyses using isolated primary calf hepatocytes from three healthy female calves (1 d old, 42 to 48 kg). Two hours prior to an FA challenge (1.2 mM mix), the hepatocytes were treated with 100 μM (M1), 50 μM (M2), or 25 μM (M3) of myricetin. Subsequently, hepatocytes from each donor were challenged with or without FA for 12 h in an attempt to induce metabolic stress. Data from calf hepatocyte treatment comparisons were assessed using two-way repeated-measures (RM) ANOVA with subsequent Bonferroni correction. The data revealed that hepatocytes challenged with FA had greater concentrations of TAG and nonesterified fatty acids (NEFA), oxidative stress-related MDA and H_2_O_2_, and mRNA and protein abundance of lipid synthesis-related SREBF1 and inflammatory-related NF-κB. In addition, the mRNA abundance of the lipid synthesis-related genes FASN, DGAT1, DGAT2, and ACC1; endoplasmic reticulum stress-related GRP79 and PERK; and inflammatory-related TNF-α also were upregulated. In contrast, the activity of antioxidant SOD (*p* < 0.01) and concentrations of GSH (*p* < 0.05), and the protein abundance of mitochondrial FA oxidation-related CPT1A, were markedly lower. Compared with FA challenge, 50 and 100 μM myricetin led to lower concentrations of TAG, NEFA, MDA, and H_2_O_2_, as well as mRNA and protein abundance of SREBF1, DGAT1, GRP78, and NF-κB. In contrast, the activity of SOD (*p* < 0.01) and mRNA and protein abundance of CPT1A were markedly greater. Overall, the results suggest that myricetin could enhance the antioxidant capacity and reduce lipotoxicity, endoplasmic reticulum stress, and inflammation. All of these effects can help reduce TAG accumulation in hepatocytes.

## 1. Introduction

Fatty liver is the main energy metabolism-related disease of dairy cows during the peripartal period [1]. Insufficient dry matter intake (DMI) after calving causes a state of negative energy balance (NEB) [2] and depending on the severity of NEB, high concentrations of fatty acids (FA) produced from adipose tissue mobilization often cause triacylglycerol (TAG) accumulation in the liver [3,4]. Besides TAG, high concentrations of FA can increase the content of reactive oxygen species (ROS) in hepatocytes and contribute to oxidative stress, endoplasmic reticulum stress, and induce inflammation [5,6]. Thus, enhancing the antioxidant defense system at the level of the liver could help prevent liver damage induced by a high concentration of FA.

Myricetin, flavonoid class of polyphenolic compounds, has biological activities including anti-oxidant and anti-inflammatory properties [7,8]. Previous studies have reported that myricetin can reduce high-fat diet-induced hepatic steatosis in mice [9]. Myricetin can effectively prevent the depletion of reduced glutathione (GSH) and the increase in malondialdehyde (MDA) in the liver, while reducing the TAG concentration and liver steatosis [10]. Whether myricetin can elicit similar beneficial effects in the bovine liver during a challenge with FA is unknown. The hypothesis of the present study was that the supply of myricetin could elicit a protective effect during FA challenge in vitro. The specific objective was to use isolated primary calf hepatocytes to evaluate biological responses during challenge with FA.

## 2. Materials and Methods

All of the animal protocols were approved by the Ethics Committee for Animal Care and Use of Heilongjiang Bayi Agricultural University (DWKJXY2021031, Daqing, China).

### 2.1. Isolation and Culture of Primary Hepatocytes

Three newborn healthy female Holstein calves (1 d old, 42 to 48 kg) were anesthetized with thiamylal sodium (50 mg/kg) followed by intravenous heparin (1500 IU/Kg) injection. Primary hepatocytes were isolated using a two-step collagenase perfusion method as previously described [6]. Briefly, the caudate lobe was isolated and washed with perfusion fluid A (140 mM NaCl, 10 mM HEPES, 6.7 mM KCl, 0.5 mM EDTA, and 2.5 mM glucose, pH 7.4, 37 °C). The perfusion solution A was then used for perfusion at a flow rate of 50 mL/min for 12 min. Subsequently, the liver was perfused with perfusion fluid B (140 mM NaCl, 30 mM HEPES, 6.7 mM KCl, 5 mM CaCl_2_, and 2.5 mM glucose, pH 7.4, 37 °C) at a rate of 50 mL/min for 5 min. Lastly, the liver was then perfused with collagenase IV (Type IV Collagenase, Gibco, Grand Island, NY, USA) solution (perfusion solution B with 0.0002 g/mL collagenase IV, pH 7.4, 37 °C) at a flow rate of 20 mL/min until the liquid became turbid. Then, 50 mL of fetal bovine serum (FBS; Hyclone Laboratories, Logan, UT, USA) was added to terminate the digestion. Upon removal of the liver capsule, blood vessels, fat, and connective tissue, the shredded liver tissue was resuspended in 4 °C pre-cooled RPMI1640 medium and filtered sequentially with 100 mesh (150 μm) and 200 mesh (75 μm) cell sieves. The filtered hepatocyte suspension was washed twice with an RPMI-1640 basal medium by centrifugation at 500× *g* for 5 min at 4 °C. Primary hepatocytes were resuspended in an adherent medium (RPMI-1640, 10% FBS, 1% penicillin Streptomyces Hepatocytes, 10^−6^ mol/L insulin, 10^−6^ mol/L dexamethasone, and 10 μg/mL vitamin C) and seeded into six-well tissue culture plates at a density of 1 × 10^6^ cells/mL and incubated at 37 °C in a 5% CO_2_ incubator for 4 h. The medium was replaced with growth medium (RPMI-1640 with 10% FBS), and the growth medium was replaced every 24 h. After 44 h of culture, the cells were starved in a serum-free RPMI-1640 basal medium for another 12 h for subsequent studies.

### 2.2. FA Preparation

Stock solution was prepared by diluting the individual FA in 0.1 M KOH at 60 °C and pH adjusted to 7.4 with 1 M hydrochloric acid. The stock FA (52.7 mM) solution contained oleic (22.9 mM; Sigma-Aldrich, St. Louis, MO, USA), linoleic (2.6 mM; Sigma-Aldrich), palmitic (16.8 mM; Sigma-Aldrich), stearic (7.6 mM; Sigma-Aldrich), and palmitoleic acid (2.8 mM; Sigma-Aldrich). This ratio and concentrations were based on previous studies on the FA composition of ketosis cows and other in vitro studies [11,12].

### 2.3. Hepatocytes Treatment

After starvation for 12 h, hepatocytes (n = 6 replicates per group; 3 calves and 2 technical replicates per group) were divided into eight groups: control, FA, M1 (100 μM of myricetin), FA + M1, M2 (50 μM of myricetin), FA + M2, M3 (25 μM of myricetin), and FA + M3. For the control and FA groups, hepatocytes were maintained in RPMI-1640 basal medium containing 2% BSA, and were treated with or without 1.2 mM FA for 12 h. For the M1, FA + M1, M2, FA + M2, M3, and FA + M3 groups, 100 μM (M1), 50 μM (M2) or 25 μM (M3) of myricetin (M6760, St. Louis, MO, USA) was used for 2 h prior to incubation with or without FA.

To evaluate the antioxidant capacity of myricetin, after starvation for 12 h, the cells were treated with 10 mM of the oxidative stress inhibitor N-acetylcysteine (NAC, Beyotime Biotechnology, Nantong, China) for 2 h prior to culture with or without FA.

### 2.4. Mesurement of TAG and NEFA Content

The TAG and nonesterified fatty acid (NEFA) contents were measured using an enzymatic TAG kit (E1013, Applygen Technologies Inc., Beijing, China) and NEFA kit (A042-2, Jiancheng Biological Technology Co., Ltd., Nanjing, China), respectively. Briefly, the collected hepatocytes were homogenized in 0.1 mL of lysis buffer, the samples were centrifuged at 14,000× *g* for 5 min, and the supernatant was used to determine TAG and NEFA contents according to the manufacturer’s instructions. Primary hepatocyte supernatant also was used to determine the protein content using the BCA assay (Beyotime Biotechnology, Nantong, China) to normalize the TAG and NEFA contents.

### 2.5. Immunofluorescence of Lipid Droplets

Approximately 5000 cells per well were inoculated into a 12-well culture plate. A 1-mL cell suspension was added to each well and treated with or without Myricetin or NAC, and then incubated with or without 1.2 mM fatty acids. For lipid droplet fluorescence, hepatocytes were washed with PBS and fixed at room temperature with 4% paraformaldehyde for 30 min. Cellular staining was performed using BODIPY 493/503 (Invitrogen Corporation, Carlsbad, CA, USA). The nucleus was stained with Hoechst (Beyotime Biotechnology, China) at room temperature for 8 min. Fluorescence was determined using an inverted microscope (IX73, Olympus, Tokyo, Japan).

### 2.6. Determination of Hepatocyte Oxidation and Anti-Oxidant Indices

The content of glutathione (GSH), malondialdehyde (MDA), and hydrogen peroxide (H_2_O_2_), and the activity of superoxide dismutase (SOD) were determined using commercial kits (Micro-Reduced GSH Test Kit (A006-1-1, Nanjing Jiancheng Bioengineering Institute, Nanjing, China), MDA Assay Kit (S0131S, Beyotime, Shanghai, China), Total SOD Assay Kit (S0101, Beyotime, Shanghai, China), and H_2_O_2_ Assay Kit (S0038, Beyotime, Shanghai, China)). Briefly, the collected hepatocytes were homogenized at 4 °C temperatures using the lysis solution of the kits. Homogenates were then centrifuged at 12,000× *g* for 10 min, and the supernatants were used to determine the MDA, GSH, and H_2_O_2_ content and SOD activities. Primary hepatocyte supernatant was also used to determine the protein content using the BCA Protein Assay kit (Beyotime Biotechnology, Shanghai, China) to normalize the MDA, GSH, and H_2_O_2_ contents and SOD activity.

### 2.7. Protein Extraction and Western Blotting

The total protein was lysed using a radioimmunoprecipitation assay (RIPA) buffer (Beyotime Biotechnology, Nantong, China) containing protease inhibitors, and protein measured using the BCA protein assay kit (Beyotime, China). Proteins (30 μg/lane) were separated in 10% SDS-polyacrylamide gels and transferred onto polyvinylidine fluoride (PVDF) membranes. Subsequently, the PVDF membrane was blocked with 5% skim milk powder dissolved in tris-buffered saline (50 mM Tris, pH 7.6, 150 mM NaCl, and 0.1% Tween 20) at room temperature for 1 h. The blocked membranes were incubated overnight at 4 °C with specific antibodies for glucose-regulated protein 78 (GRP78, 1:250, sc-376768; Santa Cruz, CA, USA), sterol regulatory element-binding transcription factor 1 (SREBF1, 1:1000, NB100-2215; Novus, St. Charles, MO, USA), carnitine palmitoyl transferase 1A (CPT1A, 1 μg/mL, ab83862, Abcam, Cambridge, MA, USA), diacylglycerol acyltransferase 1 (DGAT1, 1:1000, ab122924, Abcam, Cambridge, MA, USA), nuclear factor kappa B (NF-κB) p65 antibody (1:1000, 8242, Cell signaling, Danvers, MA, USA), and β-actin (1:1000, sc-47778; Santa Cruz, CA, USA), which we validated previously [13,14]. Subsequently, the membranes were washed and incubated with horseradish peroxidase-conjugated secondary antibodies (3:5000; Beyotime) for 45 min at room temperature. Protein abundance was visualized via an enhanced chemiluminescence solution (Beyotime) using a ProteinSimple imager (ProteinSimple, San Jose, CA, USA). The band intensity was quantified using Image Lab software (Bio-Rad, Hercules, CA, USA). The target protein abundance was normalized to the β-actin abundance.

### 2.8. RNA Extraction and Quantitative Real-Time PCR

The total RNA was extracted from the hepatocytes using TRIzol (Invitrogen Corporation, Carlsbad, CA, USA) according to the manufacturer’s protocol. The concentration of purified RNA was determined with a K5500 microspectrophotometer (Beijing Kaiao Technology Development Ltd., Beijing, China). Then, 1 μg of total RNA was transcribed into cDNA via an oligonucleotide primer using Reverse Transcriptase M-MLV (RNase H-) (RR047A, TaKaRa Biotechnology Co., Ltd., Dalian, China). The qRT-PCR reaction mixture contained 2 µL of cDNA, 1 uM of each primer, 10 µL of 2 × inNova Taq SYBR^®^ Green qPCR PreMix (SQ121, Innova, Changsha, China), and sterile water, with a final volume of 20 µL. qRT-PCR was performed on a BioRad iCycler iQTM Real-Time PCR Detection System (Bio-Rad Laboratories Inc., Hercules, CA, USA). The reaction conditions were as follows: initial denaturation at 95 °C for 3 min, followed by 40 cycles of 95 °C for 15 s, 60 °C for 1 min, and 72 °C for 30 s, and extension at 72 °C for 5 min. The relative mRNA abundance of SREBF1, fatty acid synthase (FASN), acetyl coenzyme A carboxylase 1 (ACC1), DGAT1, diacylglycerol acyltransferase 2 (DGAT2), CPT1A, GRP78, pancreatic endoplasmic reticulum eukaryotic translation initiation factor 2 kinase (PERK), NF-κB, and tumor necrosis factor α (TNF-α) was normalized to the geometric mean of β-actin and glyceraldehyde-3-phosphate dehydrogenase (GAPDH). The 2^–ΔΔCT^ method was used to determine the treatment effects on abundance. The gene primers were from our previous work [14,15] or were designed using Applied Biosystems Primer Express software, and are shown in Table 1.

### 2.9. Statistical Analysis

The data were analyzed with the Statistical Package for the Social Sciences (SPSS) 26.0 software (SPSS Inc., Chicago, IL, USA) and were presented with the GraphPad prism 8.0 software (GraphPad Software Inc., La Jolla, CA, USA). The results are reported as means ± standard error of the means. All data were tested for homogeneity of variance. Statistical significance between the control and FA, M1, M2, M3, or NAC, and between FA and FA + M1, FA + M2, FA + M3, or FA + NAC, was evaluated using two-way repeated-measure (RM) ANOVA along with the Bonferroni procedure to control for a type I error rate at 0.05. A *p*-value < 0.05 was considered statistically significant and a *p*-value < 0.01 was considered highly significant.

## 3. Results

### 3.1. Effects of Myricetin on Lipid Accumulation

Compared with the control, the concentrations of TAG and NEFA were greater with 1.2 mM of FA treatment (*p* < 0.01; Figure 1A). However, hepatocytes TAG and NEFA with a concentration of 100 μM (M1), 50 μM (M2), or 25 μM (M3) of myricetin treatment had no statistical significance compared with the control (*p* > 0.05; Figure 1A). Compared with the FA group, the concentration of TAG was lower in the FA + M1 (*p* < 0.05) and FA + M2 (*p* < 0.01) treatment groups, and the concentration of NEFA was lower in the FA + M1 (*p* < 0.01) and FA + M2 (*p* < 0.02) groups. However, the concentration of TAG and NEFA in the FA + M3 group had no significant difference compared with the control (*p* > 0.05; Figure 1A). The accumulation of lipid droplets in the hepatocytes was consistent with the changes observed in the TAG concentration (Figure 1B). These results show that 50 μM and 100 μM of myricetin could reduce the accumulation of TAG and NEFA in hepatocytes induced by high fatty acids.

### 3.2. Effects of Myricetin on Lipid Metabolism

Compared with the control, the mRNA and protein abundance of lipid synthesis-related SREBF1, and mRNA abundance of lipid synthesis-related FASN, DGAT1, DGAT2, and ACC1 were greater in the 1.2 mM FA treatment (*p* < 0.01; Figure 2B), while the mRNA (*p* < 0.05) and protein (*p* < 0.01) abundance of the mitochondrial FA oxidation-related CPT1A were lower (Figure 2B). Compared with the control, the M1 and M2 treatment groups caused a lower protein abundance of SREBF1 and DGAT1 (*p* < 0.01; Figure 2B). The protein abundance of SREBF1 (*p* < 0.01) and DGAT1 (*p* < 0.05) in the M3 treatment group was lower compared with the control (Figure 2B).

Compared with the FA group, the FA + M1 treatment group had a lower mRNA and protein abundance of SREBF1 (*p* < 0.05 and *p* < 0.01, respectively) and DGAT1 (*p* < 0.01), and a lower mRNA abundance of ACC1 and FASN (*p* < 0.01; Figure 2B). However, the mRNA (*p* < 0.05) and protein (*p* < 0.01) abundance of CPT1A in the FA + M1 treatment group were greater compared with the FA (Figure 2B). The FA + M2 treatment group had a lower mRNA and protein abundance of SREBF1 (*p* < 0.05 and *p* < 0.01, respectively) and DGAT1 (*p* < 0.01), and a lower mRNA abundance of DGAT2 (*p* < 0.05), ACC1 (*p* < 0.05), and FASN (*p* < 0.01; Figure 2B). The mRNA and protein abundance of CPT1A in the FA + M2 treatment group were greater compared with the FA (*p* < 0.01; Figure 2B). However, the FA + M3 treatment group only led to lower mRNA and protein abundance of SREBF1 and DGAT1 (*p* < 0.01; Figure 2B).

### 3.3. Effect of Myricetin on the Oxidative and Antioxidant Indexes of Hepatocytes

Compared with the control group, the concentration of oxidative stress-related MDA (*p* < 0.01; Figure 3A) and H_2_O_2_ (*p* < 0.01; Figure 3B) was greater with the 1.2 mM FA treatment. However, the activity of antioxidant-related SOD (*p* < 0.01; Figure 3C) and the concentration of antioxidant-related GSH (*p* < 0.05; Figure 3D) was lower in the FA treatment group compared with the control group. The M1, M2 and M3 treatment groups indicated no significant differences in MDA, H_2_O_2_, and GSH contents and SOD activity compared with the control group.

Compared with the FA group, FA + M1 and FA + M2 led to lower MDA (*p* < 0.01; Figure 3A) and H_2_O_2_ (*p* < 0.01; Figure 3B) contents. The activity of SOD in the FA + M1 and FA + M2 treatment groups was greater compared with the FA group (*p* < 0.01; Figure 3C). However, FA + M3 treatment only led to a lower concentration of MDA compared with the FA group (*p* < 0.05; Figure 3A). The concentrations of GSH in the FA + M1, FA + M2, and FA + M3 treatment groups had no significant differences compared with the FA group (*p* > 0.05; Figure 3D).

### 3.4. Effects of Myricetin on Endoplasmic Reticulum Stress and Inflammation

Compared with the control group, the mRNA and protein abundance of inflammatory-related NF-κB (*p* < 0.01), and the mRNA abundance of endoplasmic reticulum stress-related GRP78 and PERK and inflammatory-related TNF-α were greater in the 1.2 mM FA treatment group (*p* < 0.01; Figure 4B). The myricetin treatment groups had no significant differences compared with the control group (except a lower protein abundance of GRP78 in M2 treatment group with *p* < 0.05).

Compared with the FA group, the protein and mRNA abundance of GRP78 and NF-κB, and the mRNA abundance of PERK and TNF-α in the FA + M1 and FA + M2 treatment were lower compared with the FA group (*p* < 0.05; Figure 4B). The protein and mRNA abundance of NF-κB (*p* < 0.01), and the protein abundance of GRP78 (*p* < 0.05), and the mRNA abundance of PERK (*p* < 0.05) and TNF-α (*p* < 0.05) in the FA + M1 group were lower than the FA treatment (Figure 4B), while the FA + M1 group also led to a lower protein and mRNA abundance of GRP78 (*p* < 0.01), and mRNA abundance of PERK (*p* < 0.01) and NF-κB (*p* < 0.05) compared with the FA group (Figure 4B). However, the FA + M3 treatment group only led to a lower mRNA abundance of GRP78 compared with the FA group (*p* < 0.05; Figure 4B).

### 3.5. Effects of NAC on Lipid Metabolism

Similar to the effect of myricetin on lipid metabolism, NAC + FA also led to lower concentration of TAG compared with the FA group (*p* < 0.05; Figure 5A). The fluorescence intensity of the lipid droplet in the NAC + FA treatment hepatocytes were weaker than in the FA group (Figure 5B). The mRNA abundance of DGAT1 and SREBF1 were lower in the NAC + FA treatment compared with the FA group.

## 4. Discussion

High concentrations of circulating FA cause fatty liver in dairy cows during periods of NEB [16]. Furthermore, high concentrations of FA induced TAG accumulation in the liver during NEB are also associated with reduced mitochondrial FA beta-oxidation, as well as an increased expression of cellular markers of de novo lipogenesis, endoplasmic reticulum stress, oxidative stress, and inflammation [15,17,18].

In this study, we successfully simulated the fatty liver phenotype in primary calf hepatocytes by challenging with FA: hepatocytes accumulated TAG and high concentrations of NEFA, and had a greater abundance of lipid synthesis-related (FASN, ACC1, SREBF1, and DGAT1), endoplasmic reticulum stress-related (GRP78 and PERK), inflammation-related (NF-κB and TNF-α), and oxidative stress-related (MDA and H_2_O_2_) markers. At the same time, there was a markedly lower abundance of FA oxidation-related (CPT1A) and antioxidant-related (SOD and GSH) markers. Thus, experimental conditions allowed for addressing our objectives pertaining to the “second hit” theory of non-alcoholic fatty liver, stating that high FA concentrations are lipotoxic as a result of ROS production and the development of oxidative stress damage to liver cells. These events trigger inflammatory and pro-oxidant responses [19,20] and cause TAG accumulation in hepatocytes [21]. Thus, functional research on natural ingredients with an antioxidant capacity has become an important focus in the prevention and treatment of fatty liver.

Myricetin, as a natural antioxidant, has been widely studied in non-ruminants to prevent multiple diseases [7]. In the present study, myricetin, especially 50 μM or 100 μM treatment, markedly reduced the concentrations of oxidative stress-related MDA and H_2_O_2_, and improved SOD activity in the hepatocytes challenged with FA. This suggests that myricetin plays an important role as an antioxidant, but the lack of effect in GSH concentrations indicates that the mechanism of action was unrelated to an enhanced production of cellular antioxidants. Indeed, when oxidative stress is reduced, it diverts the utilization of serine from the transsulfuration pathway towards the synthesis of phospholipids, which helps diminish the accumulation of TAG [22].

Here, 50 μM and 100 μM myricetin markedly reduced the concentrations of TAG and the abundance of FASN, ACC1, SREBF1, and DGAT1 in hepatocytes challenged with FA. However, myricetin enhanced the abundance of the FA oxidation-related protein CPT1A. At the same time, 50 μM and 100 μM myricetin significantly down-regulated the protein abundance of GRP78, and the mRNA abundance of PERK and NF-κB in hepatocytes. Within the liver, the transport of incoming FA into the mitochondria via CPT1A for β-oxidation is a key pathway that reduces the esterification of FA [14]. However, as FA oxidation generates ROS, it is essential that the cell possesses effective antioxidant responses [23]. When excessive ROS accumulate, it can inhibit the feedback of mitochondrial oxidation and promote the use of FA for TAG accumulation [24,25]. In addition, markedly reduced concentrations of NEFA in FA + M1 and FA + M2 treatment hepatocytes should be attributed to the antioxidant effects of myricetin, which promote FA mitochondrial oxidation and reduce concentrations of MDA and H_2_O_2_ [26]. Thus, myricetin could play a major role in the antioxidant capacity of the bovine liver when it is challenged with high levels of FA. To confirm this, we added the antioxidant NAC and the results revealed a reduction in TAG along with increases in the abundance of the FA oxidation-related protein CPT1A.

Although the current studies on the hepatic regulatory mechanism of FA metabolism have been mostly performed in calf hepatocytes [14,27,28], studies have also shown that lipid metabolism efficiency is related to the age of non-ruminant animals [28], and the composition of the culture medium and the culture duration also has a significantly effect on the cells’ partial nutrient metabolism [29]. Therefore, caution should be exercised when attempting to infer liver FA metabolism in adult cows from the hepatocytes of 1-day-old calves.

## 5. Conclusions

Myricetin supply in bovine hepatocytes during high FA challenge promoted antioxidant-related SOD activity and increased the abundance of FA oxidation-related CPT1A, and down-regulated the abundance of lipid synthesis-related FASN, ACC1, SREBF1, and DGAT1; endoplasmic reticulum stress-related GRP78 and PERK; and inflammation-related NF-κB and TNF-α. This suggests that myricetin supply can induce an antioxidant effect in part by promoting FA mitochondrial oxidation, hence reducing endoplasmic reticulum stress, inflammation, and lipotoxicity. As a result, lipid accumulation in the hepatocytes is prevented. In summary, the present study provides anti-oxidant and promoting FA mitochondrial oxidation effects of myricetin for hepatocytes during a high FA challenge.

## Figures and Tables

**Figure 1 metabolites-12-01071-f001:**
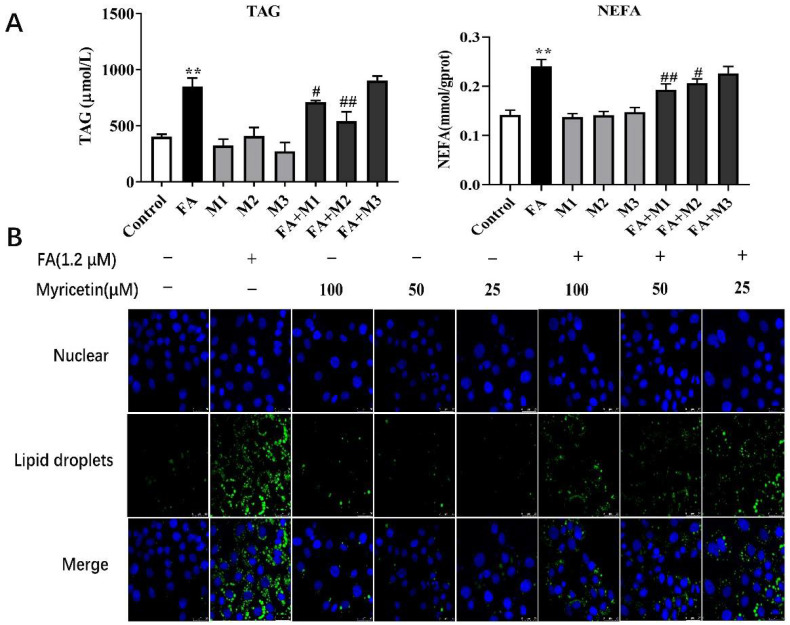
The effect of different concentrations of myricetin on TAG concentration and immunofluorescence of lipid droplets in primary calf hepatocytes. Hepatocytes were treated with 100 μM (M1), 50 μM (M2), 25 μM (M3), or RPMI-1640 basic medium (Control) with or without 1.2 mM FA. (**A**) TAG content in hepatocytes. (**B**) Immunofluorescence results of lipid droplets, scale bar = 25 um. Data were analyzed using a two-way repeated-measure (RM) ANOVA with subsequent Bonferroni correction. The data presented are the means ± SEM; ** *p* < 0.01 indicate differences from the control. # *p* < 0.05, ## *p* < 0.01 indicate differences from cells incubated with FA.

**Figure 2 metabolites-12-01071-f002:**
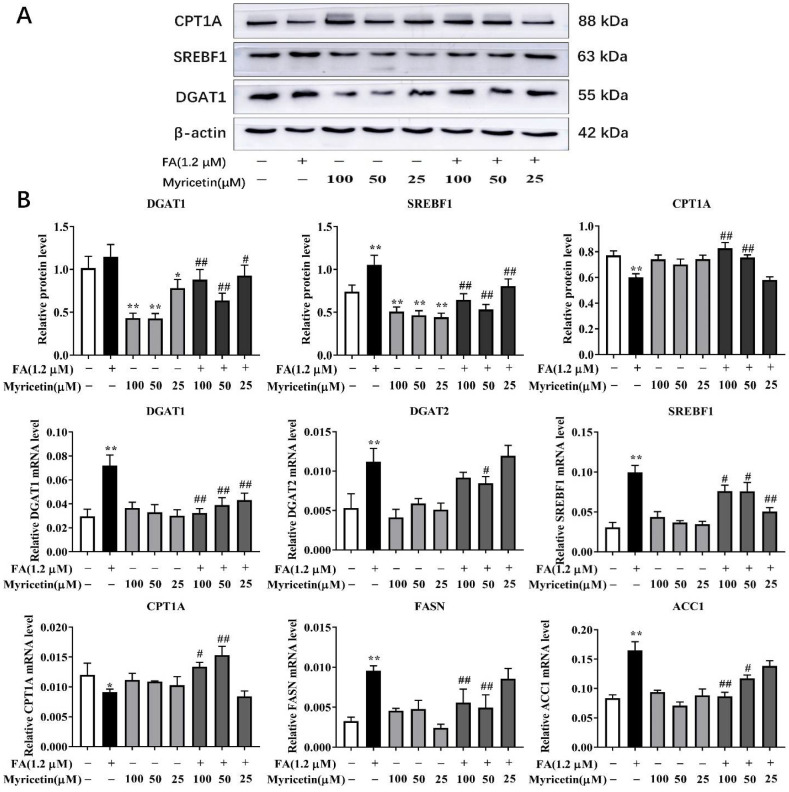
The effect of myricetin on key proteins and mRNA associated with lipid synthesis and FA oxidation. Hepatocytes were treated with 100 μM (M1), 50 μM (M2), 25 μM (M3), or RPMI-1640 basic medium (Control) with or without 1.2 mM FA. (**A**) Representative Western blot of SREBF1, DGAT1, and CPT1A. (**B**) Protein and mRNA abundance of SREBF1, DGAT1, DGAT2, CPT1A, ACC1, and FASN. Data were analyzed using a two-way repeated-measures (RM) ANOVA with subsequent Bonferroni correction. The data presented are the means ± SEM; * *p* < 0.05, ** *p* < 0.01 indicate differences from control. # *p* < 0.05, ## *p* < 0.01 indicate differences from cells incubated with FA.

**Figure 3 metabolites-12-01071-f003:**
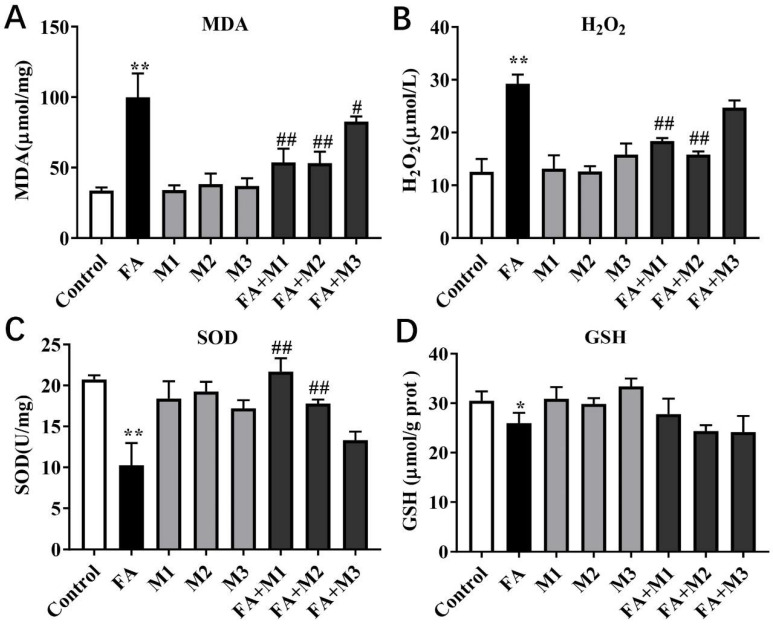
The effect of myricetin on oxidative stress in primary calf hepatocytes. Hepatocytes were treated with 100 μM (M1), 50 μM (M2), 25 μM (M3), or RPMI-1640 basic medium (Control) with or without 1.2 mM FA. (**A**) Concentration of malondialdehyde (MDA). (**B**) Concentration of hydrogen peroxide (H_2_O_2_). (**C**) Activity of superoxide dismutase (SOD). (**D**) Concentration of glutathione (GSH; gprot = g protein). Data were analyzed using a two-way repeated-measure (RM) ANOVA with subsequent Bonferroni correction. The data presented are the means ± SEM; * *p* < 0.05, ** *p* < 0.01 indicate differences from control. # *p* < 0.05, ## *p* < 0.01 indicate differences from the cells incubated with FA.

**Figure 4 metabolites-12-01071-f004:**
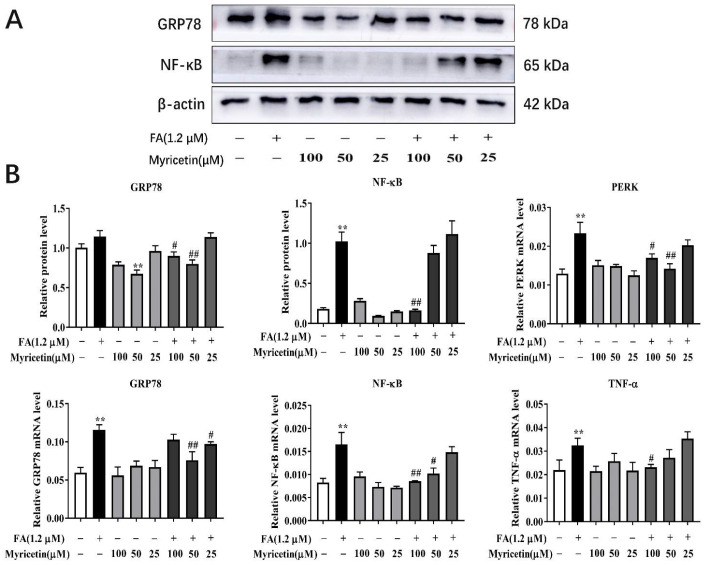
The effect of myricetin on the key protein and mRNA associated with endoplasmic reticulum stress and inflammation. Hepatocytes were treated with 100 μM (M1), 50 μM (M2), 25 μM (M3), or RPMI-1640 basic medium (Control) with or without 1.2 mM FA. (**A**) Representative Western blot of GRP78 and NF-κB. (**B**) Protein abundance of GRP78 and NF-κB, and relative mRNA abundance of PERK, GRP78, NF-κB, and TFN-α. Data were analyzed using a two-way repeated-measure (RM) ANOVA with subsequent Bonferroni correction. The data presented are the means ± SEM; ** *p* < 0.01 indicate differences from the control. # *p* < 0.05, ## *p* < 0.01 indicate differences from the cells incubated with FA.

**Figure 5 metabolites-12-01071-f005:**
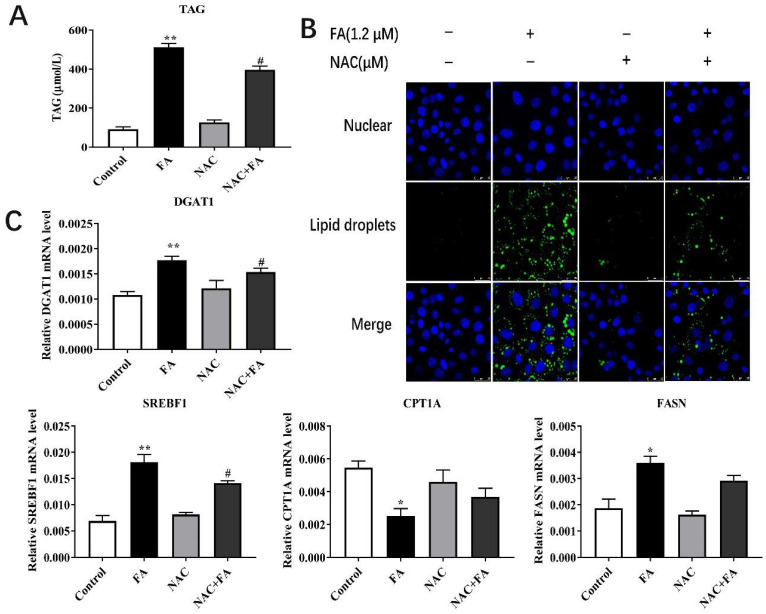
The effect of NAC on TAG concentration, lipid droplet immunofluorescence, key mRNA associated with lipid synthesis, FA oxidation in primary calf hepatocytes. Hepatocytes were treated with 10 mM NAC or RPMI-1640 basic medium (Control) with or without 1.2 mM FA. (**A**) TAG content in hepatocytes. (**B**) Immunofluorescence results of lipid droplets, scale bar = 25 μm. (**C**) Relative mRNA abundance of DGAT1, SREBP-1C, CPT1A, and FASN. Data were analyzed using a two-way repeated-measure (RM) ANOVA with subsequent Bonferroni correction. The data presented are the means ± SEM; * *p* < 0.05, ** *p* < 0.01 indicate differences from the control. # *p* < 0.05 indicate differences from cells incubated with FA.

**Table 1 metabolites-12-01071-t001:** Sequences of primers used for real-time PCR amplification.

Gene	GeneBank Number	Primer (5′ to 3′)	Length
ACTB	NM_173979.3	F: GCTAACAGTCCGCCTAGAAGCA R: GTCATCACCATCGGCAATGAG	403 bp
GAPDH	NM_001034034.2	F: GTCTTCACTACCATGGAGAAGG R: TCATGGATGACCTTGGCCAG	197 bp
DGAT1	XM_025001414.1	F: ACGCCGTGAAGTATAACCCT R: CCAAAAATCGCTTGTCCCTT	101 bp
DGAT2	NM_205793.2	F: ACCCTCATAGCCGCCTACTC R: GCCAAGTGACAGAAAACAGGT	239 bp
SREBF1	NM_001113302.1	F: GCAGCCCATTCATCAGCCAGACC R: CGACACCACCAGCATCAACCACG	119 bp
CPT1A	NM_001304989.2	F: ACGCCGTGAAGTATAACCCT R: CCAAAAATCGCTTGTCCCTT	119 bp
FASN	NM_001012669.1	F: ACAGCCTCTTCCTGTTTGACG R: CTCTGCACGATCAGCTCGAC	144 bp
GRP78	XM_024998380.1	F: GCATCGACCTGGGTACCACCTA R: CCCTTCAGGAGTGAAAGCCACA	122 bp
PERK	XM_010810067.3	F: GCCGCTCAGCTCTCCTAGTCC R: TGGCTCTCGGATGAACTGGTCTG	165 bp
NF-κB	NM_001113302.1	F: AGGACCAACCAGACCG R: TGTCACCAGGCGAGTTAT	204 bp
TNF-α	NM_001205408.1	F: CTGCCGGACTACCTGGACTAT R: CCTCACTTCCCTACATCCCTAA	144 bp

F, forward; R, reverse.

## Data Availability

Not applicable.

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
