# Peer review of "Effect of Myricetin on Lipid Metabolism in Primary Calf Hepatocytes Challenged with Long-Chain Fatty Acids"

_metabolites, 2022, doi:10.3390/metabo12111071_

Round 1
Reviewer 1 Report
This study investigated whether myricetin, a flavonoid compound, could prevent lipid accumulation in cultured primary hepatocytes. The results suggested that myricetin could enhance antioxidant capacity and reduce lipotoxicity, endoplasmic reticulum stress, and inflammation. Basically, these studies are not innovative; moreover, the authors need to address a few issues to enhance their work.
The statistical analysis of data used in the publication should be revised. Instead of one-way ANOVA, two-way ANOVA with RM should have been applied. There are the following factors: control hepatocytes vs. FA-treated in four variants (M doses): M1 (100), M2 (50), M3 (25), and M4 (0).
I suggest an explanation for why hepatocytes were collected from one-day-old calves. The fatty liver issue affects adult dairy cows that develop a negative energy balance during the postpartum period due to the sudden energy demand of lactation. Have you tried to obtain a primary cell culture of adult bovine hepatocytes?
Please remember to enter abbreviations in the text. E.g., DMI is not explained in the first paragraph of the introduction.
Author Response
Response to Comments and Suggestions
Point 1: The statistical analysis of data used in the publication should be revised. Instead of one-way ANOVA, two-way ANOVA with RM should have been applied. There are the following factors: control hepatocytes vs. FA-treated in four variants (M doses): M1 (100), M2 (50), M3 (25), and M4 (0).
Response 1: We appreciate the comment. We have modified and analyzed the significance of data using tow-way repeated-measures (RM) ANOVA when there were 2 treatment factors (FA and M doses; FA and NAC). Related figures and the description of the significance of the results were also modified.
Point 2: I suggest an explanation for why hepatocytes were collected from one-day-old calves. The fatty liver issue affects adult dairy cows that develop a negative energy balance during the postpartum period due to the sudden energy demand of lactation. Have you tried to obtain a primary cell culture of adult bovine hepatocytes?
Response 2: We appreciate the comments. Although there is no report in ruminants, we have to admit that lipid metabolism efficiency is related to age in non-ruminant animals (Hahn et al., 2017). In fact, it is difficult to collect enough liver from cows that culled because of negative energy balance during the postpartum period for primary hepatocytes culture, and the duration of warm ischemia of primary adult cow hepatocytes from abattoir-derived liver had a positive effect on cell viability, and the composition of the culture medium and the culture duration also had a significantly effect on the cells partial nutrient metabolism (Witte et al., 2019). However, a number of papers (even going back to Donkin and Armentano in the late 1980s early 1990s) have been published in JDS or other journals detailing use of calf hepatocytes to study relevant aspects of metabolism in adult cows (e.g. Graulet et al., 1998, Zhou et al., 2018, Shen et al., 2003). To make the reader aware of potential issues in extrapolating our data to in vivo conditions in adult cows we are described in detail in the last paragraph of the discussion section.
References mentioned as below:
Hahn, O.; Grönke, S.; Stubbs, T.M.; Ficz, G; Hendrich, O.; Krueger, F.; Andrews, S.; Zhang, Q.; Wakelam, M.J.; Beyer, A.; Reik, W.; Partridge, L. Dietary restriction protects from age-associated DNA methylation and induces epigenetic reprogramming of lipid metabolism. Genome Biol 2017, 18, 56,
Graulet, B.; Gruffat, D.; Durand, D; Bauchart, D. Fatty acid metabolism and very low density lipoprotein secretion in liver slices from rats and preruminant calves. J Biochem 1998, 124, 1212-1219.
Zhou, Y.F.; Zhou, Z.; Batistel, F.; Martinez-Cortes, I.; Pate, R.T.; Luchini, D.L.; Loor, J.J. Methionine and choline supply alter transmethylation, transsulfuration, and cytidine 5'-diphosphocholine pathways to different extents in isolated primary liver cells from dairy cows. J Dairy Sci 2018, 101, 11384-11395.
Shen, T.; Li, X.; Jin, B.; Loor, J.J.; Aboragah, A.; Ju, L.; Fang, Z.; Yu, H.; Chen, M.; Zhu, Y.; Ouyang, H.; Song, Y.; Wang, Z.; Du, X.; Liu, G. Free fatty acids impair autophagic activity and activate nuclear factor kappa B signaling and NLR family pyrin domain containing 3 inflammasome in calf hepatocytes. J Dairy Sci 2021, 104, 11973-11982.
Witte, S.; Brockelmann, Y.; Haeger, J.D.; Schmicke, M. Establishing a model of primary bovine hepatocytes with responsive growth hormone receptor expression. J Dairy Sci 2019, 102, 7522-7535.
Point 3: Please remember to enter abbreviations in the text. E.g., DMI is not explained in the first paragraph of the introduction.
Response 3: Thanks for the comment. We have revised in line 43, 128 and 184 and highlighted.

Reviewer 2 Report
The study examines the in vitro analyses of antioxidant capacity and reduction in lipotoxicity, endoplasmic reticulum stress, and inflammation using myricetin in isolated primary calf hepatocytes from healthy female calves. There are a few minor issues that need to be addressed such as:
Page 1; Line 24-26: The result should be constructed in a way that all the activities outcomes are represented either in the percentage or statistically significant values.
Page 1; Line 29-30: The result should be constructed in a way that all the activities outcomes are represented either in the percentage or statistically significant values.
Page 2; Line 18: It would be great if the Animal Ethics approval number should be given here in the section.
Page 3; Line 25: Need to brief the method "Determination of Hepatocyte Oxidation and Anti-oxidant Indices" even using commercial manufacturing kit.
The results of each section need to be elaborate more than the written paragraphs which are quite briefly explained.

Author Response
Response to Comments and Suggestions
Point 1: Page 1; Line 24-26: The result should be constructed in a way that all the activities outcomes are represented either in the percentage or statistically significant values.
Page 1; Line 29-30: The result should be constructed in a way that all the activities outcomes are represented either in the percentage or statistically significant values.
Response 3: We appreciate the comment. We have added the statistically significant values for activities outcomes in line 30 and line 33.
Point 2: Page 2; Line 18: It would be great if the Animal Ethics approval number should be given here in the section.
Response 2: Thanks for the comment. We have added the Animal Ethics approval number in Page 2; Line 18.
Point 3: Page 3; Line 25: Need to brief the method "Determination of Hepatocyte Oxidation and Anti-oxidant Indices" even using commercial manufacturing kit.
Response 3: Thanks for the comment. We have added the briefly described in Page 3; Line 31.
Point 4: The results of each section need to be elaborate more than the written paragraphs which are quite briefly explained.
Response 4: Thanks for the comment. We have elaborated explained the results in detail of each section and highlighted.

Reviewer 3 Report
This an original paper focused on estimation of effect of myricetin on lipid metabolism in primary calf hepatocytes challenged with long-chain fatty acids. Subject of article is current and important because myricetin supplementation could strengthen the antioxidant defense system and reduce inflammatory responses at the level of the liver, which could help prevent liver damage induced by high concentration of FA in cattle. However, I also have a few reservations about the manuscript, which are listed below.
Comments:
Materials and Methods:
- Page 4 - the authors used the qRT-PCR reaction FastStart Universal SYBR Green Master (La Roche), which typically requires polymerase activation at 95°C for 10 minutes for 100% efficiency (the authors report 3 minutes), it is unclear why the authors modified the standard procedure.
Results:
- Page 5 - it is not clear from Figure 1A that there was a demonstrably lower concentration of TAG in the M2 treatment compared to the control. At the same time, Figure 1A lacks significance above the columns for M1 and M3.
- Page 5, line 24 – Figure 2B.
- Page 6 - Figure 2 - remove "C" from the figure - only panels A and B are in the description below the figure.
- Page 6 - Figure 2 - in the text of the results for the comparison of the FA and FA+myricetin groups, which are subsequently presented by the figure, the significances are at the P<0.05 level however the figure shows that these significances are at the P<0.01 level (protein abundance of DGAT1, SREBF1 and mRNA abundance of DGAT1, SREBF1, FASN, ACC1)
- Page 7 - Figure 3 - Similarly in the text of the results for the comparison of FA and FA+myricetin groups, which are subsequently presented by the figure, the significances are at the P<0.05 level however the figure shows that these significances are at the P<0.01 level (MDA, H2O2). Similarly, for the comparison of the control and FA groups, the text shows that the significance levels are at the P<0.05 level, but the figure shows that these significance levels are at the P<0.01 level (SOD)
- Page 7 - missing description below figure 3
- Page 8 – figure 4 – in the figure keep only the marking of panels A and B (remove markings C, D, E, F, G). As in the case of Figures 2 and 3, different significances in the text than those shown in the figures are reported - for the comparison of the control and FA groups (protein abundance of NFkB and mRNA abundance of GRP78, PERK, NFkB, TNFa) and for the comparison of the FA+myricetin groups (protein abundance of NFkB and mRNA abundance of GRP78, PERK, NFkB, TNFa). The figure shows that a decrease in mRNA abundance of GRP78 and PERK occurred also in the FA+M3 treatment compared with the FA group, which is not shown in the text.
Author Response
Response to Comments and Suggestions
Materials and Methods:
Point 1: Page 4 - the authors used the qRT-PCR reaction FastStart Universal SYBR Green Master (La Roche), which typically requires polymerase activation at 95°C for 10 minutes for 100% efficiency (the authors report 3 minutes), it is unclear why the authors modified the standard procedure.
Response 1: Thanks for the comment. We used 2 × inNova Taq SYBR® Green qPCR instead of FastStart Universal SYBR Green Master (La Roche). We are sorry for the mistake. We have revised in Page 4.
Results:
Point 2: Page 5 - it is not clear from Figure 1A that there was a demonstrably lower concentration of TAG in the M2 treatment compared to the control. At the same time, Figure 1A lacks significance above the columns for M1 and M3.
Response 2: We are sorry for the mistake. TAG concentration in M1, M2 and M3 groups had no statistical significance compared with the control. We have revised in Page 5. In addition, we have revaluated the data using tow-way repeated-measures (RM) ANOVA when there were 2 treatment factors (FA and M doses; FA and NAC). Related figures and the description of the significance of the results were also modified.
Point 3: Page 5, line 24 – Figure 2B.
- Page 6 - Figure 2 - remove "C" from the figure - only panels A and B are in the description below the figure.
Response 3: Thanks for the comment. We have modified Figure 2B in the result description and removed the "C" in Figure 2.
Point 4: Page 6 - Figure 2 - in the text of the results for the comparison of the FA and FA+myricetin groups, which are subsequently presented by the figure, the significances are at the P<0.05 level however the figure shows that these significances are at the P<0.01 level (protein abundance of DGAT1, SREBF1 and mRNA abundance of DGAT1, SREBF1, FASN, ACC1)
- Page 7 - Figure 3 - Similarly in the text of the results for the comparison of FA and FA+myricetin groups, which are subsequently presented by the figure, the significances are at the P<0.05 level however the figure shows that these significances are at the P<0.01 level (MDA, H2O2). Similarly, for the comparison of the control and FA groups, the text shows that the significance levels are at the P<0.05 level, but the figure shows that these significance levels are at the P<0.01 level (SOD)
Response 4: Thanks for the comment. We are sorry for our sloppy description. We've revised in these results part and highlighted.
Point 5: Page 7 - missing description below figure 3.
Response 5: Thanks for the comment. We are sorry for the mistake. We have added a description below Figure 3. Please accept our sincere appreciation for your comments.
Point 6: Page 8 – figure 4 – in the figure keep only the marking of panels A and B (remove markings C, D, E, F, G). As in the case of Figures 2 and 3, different significances in the text than those shown in the figures are reported - for the comparison of the control and FA groups (protein abundance of NFkB and mRNA abundance of GRP78, PERK, NFkB, TNFa) and for the comparison of the FA+myricetin groups (protein abundance of NFkB and mRNA abundance of GRP78, PERK, NFkB, TNFa). The figure shows that a decrease in mRNA abundance of GRP78 and PERK occurred also in the FA+M3 treatment compared with the FA group, which is not shown in the text.
Response 6: Thanks for the comment. We've revised figure 4 in Page 9 and the significance analysis is elaborated explained in Page 8.

Reviewer 4 Report
The manuscript deals with the effect of myricetin on lipid metabolism in primary calf hepatocytes challenged with long-chain fatty acids. Myricetin, a flavonoid compound (specifically a flavonol) is well-known for its biological activities e.g. anti-oxidant and anti-inflammatory properties. Additionally it has been demonstrated that it can reduce high-fat diet-induced hepatic steatosis in mice and prevent the depletion of reduced glutathione and the increase in malondialdehyde in the liver while reducing triacylglycerols concentration and liver steatosis. The aim
of this study is the investigation of such beneficial effects in the bovine liver.
Despite interesting the main weak point is the lack of individual characterization of both triacylglycerols and fatty acids, which is crucial for this kind of study. Such determination can be easily done by using HPLC and GC analyses.
Other suggestions:
Abstract. It is too long! It has to be more concise and factual highlighting the purpose of the research, the main results and major conclusions.
Conclusions. They are too maigre. They should extended by focusing on the main results achieved and on-going studies.
Author Response
Response to Comments and Suggestions
Point 1: Despite interesting the main weak point is the lack of individual characterization of both triacylglycerols and fatty acids, which is crucial for this kind of study. Such determination can be easily done by using HPLC and GC analyses.
Response 1: We appreciate the comment. The nonesterified fatty acids (NEFA) content were measured using an enzymatic assay kit (A042-2, Jiancheng Biological Technology Co. Ltd., Nanjing, China). The experimental methods, results and related discussion were also added and highlighted.
Other suggestions:
Point 2: Abstract. It is too long! It has to be more concise and factual highlighting the purpose of the research, the main results and major conclusions.
Response 2: Thanks for the comment. We have revised the abstract.
Point 3: Conclusions. They are too maigre. They should extended by focusing on the main results achieved and on-going studies.
Response 3: Thanks for the comment. We have extended the conclusions.

Round 2
Reviewer 1 Report
The authors have made a good effort to improve their manuscript. Linguistic and grammatical errors remain to be corrected.
Reviewer 4 Report
The authors have adequately addressed all remarks and the improved paper can be now accepted in the present form